

# A model for new media data mining and analysis in online English teaching using long short-term memory (LSTM) network

Chen Chen[1] and Muhammad Aleem[2]

[1] Department of Language and Literature, College of Technology, Hubei Engineering University, Xiaogan, China
[2] National University of Computer and Emerging Sciences, Islamabad, Islamabad, Pakistan

## ABSTRACT

To maintain a harmonious teacher-student relationship and enable educators to gain a more insightful understanding of students' learning progress, this study collects data from learners utilizing the software through a network platform. These data are mainly formed by the user's learning characteristics, combined with the screen lighting time, built-in inertial sensor attitude, signal strength, network strength and other multi-dimensional characteristics to form the learning observation value, so as to analyze the corresponding learning state, so that teachers can carry out targeted teaching improvement. The article introduces an intelligent classification approach for learning time series, leveraging long short-term memory (LSTM) as the foundation of a deep network model. This model intelligently recognizes the learning status of students. The test results demonstrate that the proposed model achieves highly precise time series recognition using relatively straightforward features. This precision, exceeding 95%, is of significant importance for future applications in learning state recognition, aiding teachers in gaining an intelligent grasp of students' learning status.

## INTRODUCTION

With the continuous development of science and technology, new media technology has been updated and iterated drastically in recent years. As portable devices develop such as smartphones and tablets, our ways of contacting information have also increased, which has also led to great changes in our learning methods (*Holt, Ustad Figenschou & Frischlich, 2019*). The traditional offline teaching mode can no longer satisfy everyone's thirst for knowledge. The common strong rise of new media communication channels and new portable devices makes everyone's learning more convenient and efficient (*Zheng & Chen, 2020*). Under the influence of COVID-19, the proportion of online teaching has gradually increased in the past two years, and the teaching method has gradually changed from a pure offline teaching method to a combination of the two, and the proportion of online courses has gradually increased (*Liu & Zhang, 2018*).

English teaching, as a frontier subject of educational reform, has been the problem that generations of scholars have been seeking to reform teaching methods. It has always been

Corresponding author
Chen Chen, 15972210920@163.com

the goal of English teachers to ensure its progressiveness downstream (*Albiladi & Alshareef, 2019*). In traditional teaching, the teaching method is improved through multimedia, which increases students' interest in learning, changes the way of interaction and communication between teachers and students, and makes the relationship closer (*Zhang & Wang, 2017*). At the same time, several studies have been carried out on Task-based Language Teaching (TBLT) (*Crookes & Ziegler, 2021*). At the time of the rise of new media teaching, it has made contributions to improving teaching effects, promoting students' autonomous learning, and exploring new ways for teaching reform. At the same time, in these researches, they not only focuses on offline multimedia teaching but also actively combine with online teaching and carry out a considerable degree of forward-looking research (*Hasnain & Halder, 2021*). Nowadays, English teaching combined with the Internet in the new media era can greatly expand teaching resources through advanced technical means and communication media, and stimulate learning interests in all aspects through pictures, audio, and video. At the same time, the learning method of teaching students according to their aptitude can also be implemented, which greatly improves the disadvantages of traditional college English classroom teaching (*Meng-yue, Dan & Jun, 2020*).

Although the development of online new media has led to the further development of hybrid English teaching, the ecological environment of the whole new media online English teaching is not perfect, and it is urgent to reform, innovate, and explore new ideas for development (*Fei, 2020*). Teaching ecological environment refers to the environment where the teaching theme exists. In addition to the common natural physical social environment, the teaching resource environment, language environment, and psychological environment are all important factors in the whole teaching process (*Wu, 2022*). Therefore, in the development of English teaching ecology in universities, it is necessary to fully consider every important link in the environment, that is, to do a good job of teaching students. In teaching, each student and teacher has their niche. The mutual transmission and exclusion of the energy information of cooperation and competition between the two will make the niche constantly change (*Khanna et al., 2019*). The unit cannot ensure effective subject-learning development and niche balance. Teachers cannot exceed their niche to confuse students. At the same time, students should form an ecological niche relationship that promotes each other (*He & Fu, 2022*). For the hybrid English teaching mode under the new media presently, the important medium to ensure the balance of the teacher-student niche is the online teaching platform on the PC or mobile phone (*Sun, Anbarasan & Praveen Kumar, 2021*). These online platforms are the basic conditions for building the online teaching ecology. They put students in the center of the ecological position, and through the guidance of teachers, they actively learn knowledge and skills and improve their enthusiasm. However, the change in this environment often affects the teaching effect, and the subjective initiative of students plays a great role (*Wang, Zhang & Ye, 2021*). When students are separated from the ecosystem of classmates and from the partners and supervisors who work together, the progress is often slow, and the teaching quality is often reduced in the early stage of teaching. Therefore, there is an urgent need for an intelligent means and method to enable teachers to

understand students' learning content and learning status more quickly and accurately (*Yan, 2019*).

The learning process of students can be regarded as a time series. With the change of time, the operation of students in the terminal equipment will also change. Concerning English learning, students can complete corresponding learning tasks by operating different software and different interfaces on the terminal (*Stephanie & Sarno, 2019*). Therefore, students' learning behavior can be intelligently analyzed by recognizing and classifying the time-series characteristic signals of the equipment. For time series, although traditional machine learning technologies such as k-nearest neighbors (KNN) and C4.5 can complete the classification task, the methods based on probability and feature distance do not take time into account, so there may be differences in the results. HMM method has certain advantages in processing time series, but its model is relatively complex, and it is difficult to construct various matrices in the program and to reproduce the calculation process (*Pujianto, Prasetyo & Taufani, 2020*). With the continuous development of deep learning technology and computer computing power, recurrent neural network (RNN) has been favored by researchers, but its performance on longer time series is not satisfactory. Therefore, on this basis, researchers have proposed models such as LSTM and gated recurrent unit (GRU) that can remember the sequence state in the time series for a long time, to improve the accuracy of the models (*Hochreiter & Schmidhuber, 1997*).

To build a more harmonious new media online teaching ecology, aiming at the binding of teacher-student relations and the intelligent supervision of teachers over students' learning status, we use the network platform learning data we use to complete the observation of students' learning sequence through the LSTM model in deep learning, to timely supervise students' learning status and ensure students' enthusiasm for learning, And form an intelligent decision-making scheme to provide reference for teachers in the next step of online teaching, and improve the current online teaching ecological environment from the root of the teacher-student relationship niche in the online teaching ecology.

Main contributions in article:

(1) This article extracts multidimensional features by collecting data from learners and utilizing user learning features, combined with screen lighting time, posture of built-in inertial sensors, signal strength, and network strength.

(2) This article proposes an intelligent classification method for learning time series, which utilizes long short-term memory (LSTM) to intelligently recognize students' learning status.

(3) Comparing with the competive methods, our method achieve the best performance in the MSCOCO and Pascal VOC datasets.

## RELATED WORK

Students have many behaviors in class that can accurately reflect their learning state and are not easily interfered by other external factors, such as brain wave activity images, electrocardiogram and blood pressure and other physiological information. *Mukhopadhyay, Maka & Moorthy (2016)* used sensors to collect physiological signals from

various dimensions, such as heart rate, skin electrical conduction and blood pressure, to judge students' learning status (*Li et al., 2019*). Based on these data, they discussed the relationship between the changes of physiological signals and the learning state during the learning process of students, and proposed a learning state detection model based on physiological signals. *Woolf et al. (2009)* used posture analysis chair, pressure mouse and wireless skin conduction sensor to obtain students' sitting position, hand pressure change to mouse and skin conductivity change respectively, and comprehensively analyzed students' learning state and learning emotion through these data (*Bamidis, 2017*).

Recently, classroom learning state assessment based on deep learning technology has become the mainstream research method. This method mainly uses the cameras deployed in the classroom to collect classroom images in real time and judge the learning state of students by analyzing their behaviors. *Simonyan & Zisserman (2014)* first proposed a creative two-stream network, which uses two identical convolutional neural networks (CNNs), in which one network injects video frames to obtain spatial information, and the other network injects optical flow information of video to obtain temporal information. The last two networks are fused by means of average or support vector machine (SVM), and the experimental results show that SVM is the best in task state recognition (*Woolf et al., 2009*). Then, *Wang, Qiao & Tang (2015)* used an improved trajectory instead of optical flow to extract time information without changing the spatial network, and pooled the local ConvNet responses on a spatio-temporal tube centered on the trajectory. Finally, the Hsher vector is used to aggregate the whole video into a global hypervector, and linear SVM is used as a classifier to recognize the state of the person in the video (*Simonyan & Zisserman, 2014*). *Whitehill et al. (2014)* explored a method for automatic recognition of learning engagement from students' facial expressions. This method used SVM with Gabor features to detect students' learning engagement every 10 s in a video stream (*Wang, Qiao & Tang, 2015*). *Yang et al. (2018)* proposed a remote classroom learning state detection method based on facial expression recognition. The model was trained by six basic expressions defined on the JAFFE dataset (happiness, surprise, fear, sadness, anger and disgust), which were then used as students' basic learning states for experiments (*Whitehill et al., 2014*). *Hung et al. (2017)* used the facial expression analysis system to analyze learners' facial expressions. In their study, expression recognition was still based on six basic expressions, but different from them, they associated the six basic expressions with learning emotions and defined students' positive and negative states in class: happiness and surprise are classified as positive, while the other four are classified as negative (*Yang et al., 2018*). *Zhou et al. (2022)* proposed a method of identifying abnormal classroom behaviors by integrating temporal correlation, which detects students' abnormal behaviors based on time-contextual second-level recursion, and plays an auxiliary role in classroom dynamic management and learning effect evaluation (*Hung et al., 2017*). *Zhao, Zhu & Niu (2023)* proposed a frequent sequence mining algorithm and cluster analysis method for smart classroom. Through the analysis and verification of teaching video cases of this method, the differences of teaching modes of various disciplines can be obtained by cluster analysis for a specific discipline (*Zhou et al., 2022*). *Li et al. (2022)* proposed a class behavior recognition algorithm that integrates human body pose estimation and target

detection to extract and detect students' skeleton, providing data support for optimizing teaching design and implementing teaching intervention (*Zhao, Zhu & Niu, 2023*). At present, this kind of application can only detect students' behaviors, obtain and analyze the classroom behavior data of most students in the data, and lack the classroom behavior evaluation for a certain student, and the data cannot be used as the evaluation basis for all students in the classroom.

## METHOD

This section will introduce the processing method of the data used. The specific method flow is shown in Fig. 1:

(1) collect the data used by learners in the software through the network platform peripherals, and process the data into observation values;

(2) realize the classification of learners' learning sequences through the middle LSTM (long short-term memory) method of the deep neural network, carry out task identification, to understand their learning progress and learning effects;

(3) provide teachers with more intuitive data support, help them complete teaching supervision tasks, to achieve teaching feedback, and help students make their next plans.

Given the diverse nature of text data obtained from the network platform, processing this information uniformly poses a challenge, hindering its potential application in future analyses. Consequently, this article focuses solely on processing numerical data derived from new media learning. Following internal software processing, these data manifest as learning observation values, primarily shaped by user learning behaviors and encompassing multidimensional characteristics like screen lighting duration, built-in inertial sensor attitudes, signal strength, and network quality. Specifically, screen lighting duration signifies learning time, the built-in inertial sensor attitude reflects learning states, and network strength delineates the extent of network interference encountered during learning sessions.

### Data smooth filtering

Before feeding data into the LSTM model, several preprocessing techniques are employed to ensure optimal performance and meaningful analysis. Given the problem that there is much noise and severe jitter in the observed values formed by the learning time, network fluctuation, and signal intensity in the data derived by the client, this article first preprocesses the data to complete the smoothing. Moving average is a relatively simple method, but it should be combined with practice to construct a suitable smoothing method, mainly including mean average, exponential mean average and Savitzky Golay Filter (*Li et al., 2019*). In this article, the most common moving average method is adopted to preprocess the data. Its calculation method is shown in Formula (1), which predicts the next observation value by calculating the average value of multiple consecutive observations.

$$p_t = \frac{\sum_{i=1}^{n} (x_{t-i} + x_{t+i}) + x_t}{2n + 1} \tag{1}$$

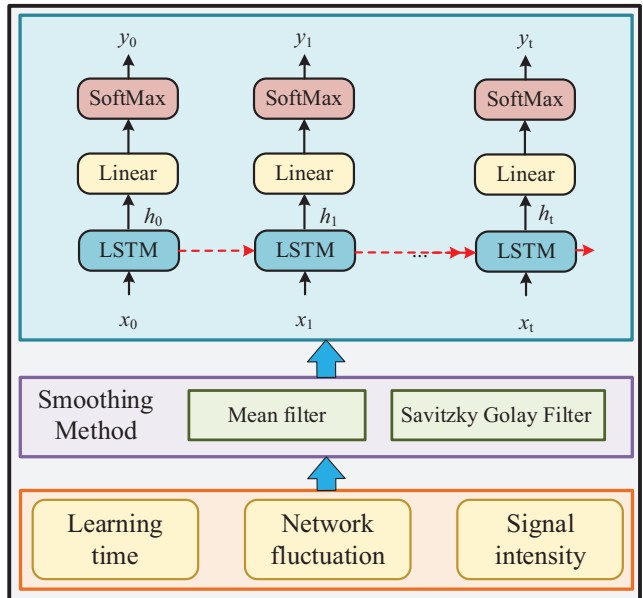

**Figure 1 Diagram of the online teaching ecology.**

where, pt represents the filtering result at time t, xt is the observation value at time t, and n represents the sliding window radius. The window size directly affects the filtering result. If the window is too large, the filtering result will be smoother, but may deviate from the true value to some extent. The smaller the window, the closer the filtering result will be to the observed value, but the noise will be larger. In addition, for different filtering requirements and considering the difference in the confidence level of observation values, the weighted moving average method is formed by adding weights to the observation values. Its calculation method is shown in Formula (2):

$$p_t = \frac{\sum_{i=1}^{n} \left( x_{t-i} * \omega_{t-i} + x_{t+i} * \omega_{t+i} \right) + x_t * \omega_t}{2n + 1} \tag{2}$$

Similar to Formula (1), where $\omega_t$ is the weight of the observation value at time t, and Formula (2) indicates that each observation value is multiplied by the weight and then averaged. This method applies to the case where the observation value itself has confidence. Since the derived data has undergone initial synthesis and is independent of each other, only commonly used smoothing filtering in Formula (1) is adopted.

## Learning status recognition based on LSTM model

The learning state of students changes with time, so the monitoring of their state needs to have the ability to continuous processing of time series. Although traditional machine learning algorithms such as KNN, SVM, LR, and so on can achieve classification in time steps, they do not have time continuity, and it is difficult to estimate the previous state, which also leads to greatly reduced accuracy of their classification and prediction results. Therefore, it is necessary to use the cyclic neural network for the sequence with time continuity (*Shi et al., 2022*). The mathematical expressions of the traditional neural

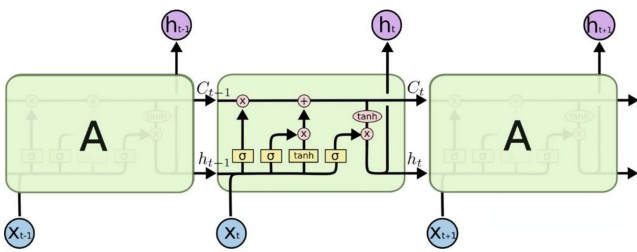

**Figure 2 Structure of the LSTM network.**

network and the recurrent neural network can be expressed by the following Formulas (3) and (4), respectively:

$$h_t = \sigma(\omega_{x_t} x_t + b) \tag{3}$$
$$h_t = \sigma(\omega_{x_t} x_t + \omega_{h_t} h_{t-1} + b) \tag{4}$$

RNN incorporates the previous time's input state, $h_{t-1}$, in its function input process. This inclusion enables the network to retain memory of the output state from the prior time, making it particularly effective in handling time series data. However, during the application of recurrent neural networks, a common issue arises known as "long-term dependence." This problem arises due to the recurrent neural network's nodes undergoing multiple iterations and calculations, which can result in the overshadowing of previous long-term features. Figure 2 illustrates the chain composition of the LSTM unit, a solution addressing this challenge, as outlined in *Yu et al. (2019)*.

It can be found that compared with the traditional RNN structure, the cell of the LSTM adds a layer of cell state at the top, through which the states at various times can be transmitted in the LSTM chain. The specific calculation process can be expressed by Eqs. (5)–(10) (*Park et al., 2020*):

$$f_t = \sigma(W_f \cdot [h_{t-1}, x_t] + b_f) \tag{5}$$
$$\widetilde{C}_t = \tanh(W_C \cdot [h_{t-1}, x_t] + b_C) \tag{6}$$
$$i_t = \sigma(W_i \cdot [h_{t-1}, x_t] + b_i) \tag{7}$$
$$C_t = f_t \times C_{t-1} + i_t \times \widetilde{C}_t \tag{8}$$
$$o_t = \sigma(W_o[h_{t-1}, x_t] + b_o) \tag{9}$$
$$h_t = o_t \times \tanh(C_t) \tag{10}$$

The LSTM process shown in Fig. 2 can be represented by the above formula. First, ft is the forgetting gate, which is the most important feature, indicating which features in Ct−1 can be used to calculate Ct. In Formula (6), $\widetilde{C}_t$ it represents the cell state update value, which is obtained by the input data and the hidden node ht−1 through the neural network layer, and updates the cell state through the tanh function. Formula (7) represents the input gate, similar to ft in Formula (5), which is also calculated by the activation function of and hidden node ht−1. After the calculation of it in Formula (7), we can determine which features can be used to update Ct. Finally, through Eqs. (9) and (10), the output ht of
| Table 1 Network architecture. | |
|---|---|
| Input size | 4 |
| LSTM layer | 64/128/256 units |
| Fully connected layer | 5 |
| Softmax layer/classification layer | / |

the hidden node can be calculated to complete the complete input and prediction value calculation of the next time unit.

Through comparative analysis, it is noteworthy that RNN offers greater convenience in processing time series compared to traditional networks due to its consideration of previous time inputs. However, when handling longer time series data to mitigate potential long-term dependency issues, LSTM emerges as an effective solution. Leveraging its gated structure, LSTM enables continuous state transfer across different time steps within the entire series. This design not only ensures a more systematic approach but also significantly enhances accuracy in handling time series data.

## Construction of the network

The construction of a neural network needs to fully consider the data characteristics and build it according to the needs. In the process of classification, the LSTM model can achieve "sequence to sequence" classification and sequence classification. According to the learning characteristics, this article intends to achieve a "sequence to sequence" classification. According to the need for mutual supervision between teachers and students in the network ecosystem under the new media English teaching, this article exports data through the network terminal to identify and classify the learning sequence. According to the needs of online teaching and teacher supervision, the learning process is divided into five categories: listening, speaking, reading, writing, and free for teacher supervision. Therefore, the form of the built network is shown in Table 1:

It can be seen from Table 1 that the input size of the built LSTM network is 4, mainly including the obtained screen light-up time, the attitude of the embedded inertial sensor, signal intensity, and network intensity, followed by the LSTM layer, and then the data passing through the LSTM layer is reduced into five categories by using the full connection layer, namely listening, speaking, reading, writing and free in English learning, that is, the full connection layer with the number of units of five. Finally, the recognition of learning state is realized through softmax function and classification layer. Because the data used in this article is not complex, only one layer of the LSTM model is established. In the process of building, to test the accuracy of the model and find the best model, this article tests the number of cells of LSTM and selects the best result. In addition, in the model algorithm of this article, LSTM introduces memory unit and gating mechanism, which can well deal with long-term dependency problems, and can easily stack multiple LSTM layers to deal with more complex sequence data. Moreover, the gating mechanism is interpretable,

which can help understand the model's processing process of sequence data. To help teachers and students themselves better understand the learning state.

## EXPERIMENTAL RESULTS AND ANALYSIS

### Data acquisition and processing

The dataset comprises anonymized student activity records within a peer review activity conducted through the Synergy platform (https://zenodo.org/records/5059711). This data pertains to an educational technology course offered at a European university and involved 30 participants hailing from various departments: Computer Science, Mathematics, Physics, Preschool Education, and Elementary Education. The primary assignment within this course centered on a learning design project where students collaborated in groups. This project spanned a duration of 3 weeks, with each project assigned 2–3 reviewers for evaluation purposes.

After the initial filtering of the collected raw data is completed by the smoothing filtering method described and the label is added, the data form is shown in Fig. 3. The input of the established model at each time is shown in Eq. (11)

$$x_t = [s_t, a_t, c_t, n_t] \tag{11}$$

where, st stands for the screen time, at stands for the attitude signal, ct stands for the signal intensity, and nt stands for the network intensity.

It can be found that for different learning methods, the observation values of students have obvious differences. To display more clearly, different positions in the sequence are marked in the figure and expressed in different colors.

### Experimental results

According to the network model built and the corresponding data collected in "Data Smooth Filtering", the data is trained and tested. Due to the lack of support from the public data set, this article can not use the common training set test set division method in deep learning and uses the one leave-out method commonly used in machine learning to complete the test. Under the test of different LSTM units, the recognition accuracy results are shown in Table 2:

According to the data in Table 2, with the increase in the number of LSTM hidden nodes, the accuracy will be improved to a certain extent, but the corresponding calculation time will also be lengthened. When the number of nodes reaches 256, the recognition effect is lower than that of the network with 128 hidden nodes. This situation has also been explained in other articles. The data we collected is relatively simple and the form is not complex, when the recognition accuracy reaches a certain level, it is impossible to continue to greatly increase the recognition accuracy of the model by changing the number of hidden nodes. Therefore, this article selects a model of 128 nodes to achieve high-precision classification and division of student learning sequences.

The comparison between the prediction results of the model established by 128 hidden nodes and the results of artificial tags is shown in Fig. 4. It can be found that there is confusion between the recognition of individual points and individual periods in the

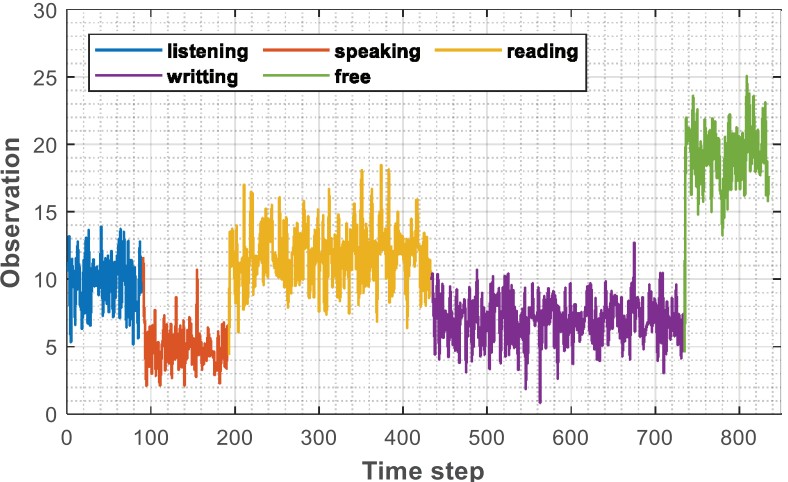

**Figure 3 Labeled sequence.**               

**Table 2 Recognition accuracy with different LSTM units.**

| Number of LSTM units | Accuracy |
| --- | --- |
| 64 | 95.96% |
| 128 | 96.29% |
| 256 | 96.10% |

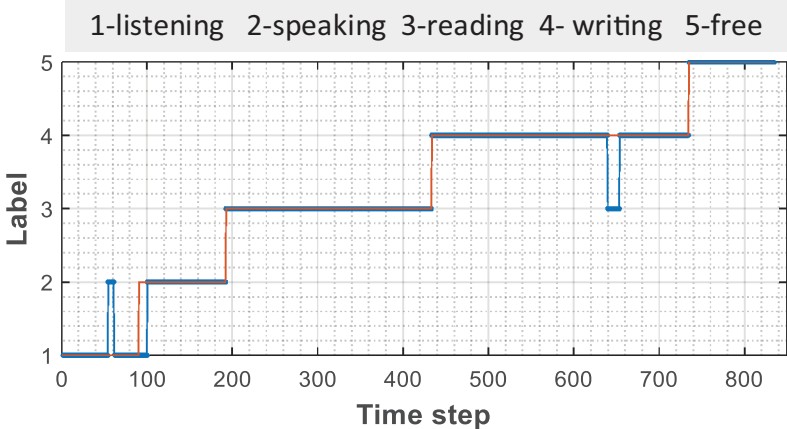

**Figure 4 The comparison between the manual and predicted annotation.**

recognition process of listening and writing, which may be caused by network fluctuations in the process of data collection or by volunteers' accident touch; however, such errors can better help teachers to observe students' learning state more intelligently from the original curve.

Figure 5 illustrates that, in comparison to other model algorithms, the LSTM method exhibits the highest accuracy, reaching 95%. This superior performance can be attributed

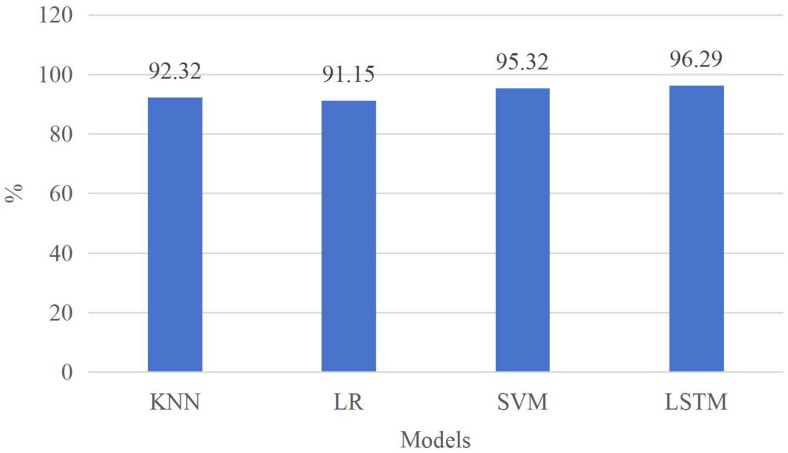

**Figure 5 Comparison with other classification methods.**

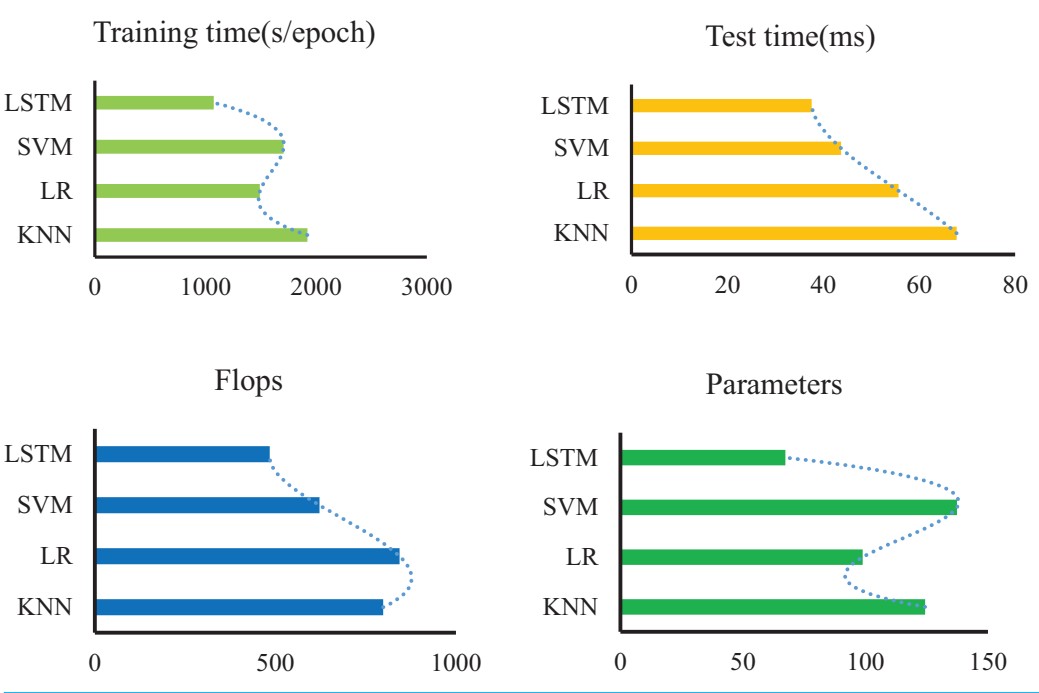

**Figure 6 Comparison with other methods in terms of time, flops and parameters.**

not only to the utilization of average sliding processing during data preprocessing, which effectively filters out redundant data noise but also to the inherent suitability of the LSTM model for processing sequential data. Consequently, in accordance with Formula (11), the identification of inputs can be executed more efficiently leveraging these four types of input features. This underscores the LSTM method's capability to achieve precise recognition of time series utilizing relatively straightforward features, a crucial aspect in future learning state recognition applications. Moreover, this approach furnishes educators with a superior foundation to comprehend students' statuses within the teaching environment.

Furthermore, our method's performance is validated through a comparative analysis involving KNN (*Hu et al., 2016*), LR (*Liu, Yin & Hu, 2020*), and SVM (*Wang & Zhong, 2003*). The LSTM algorithm showcases notable metrics: a single-round training time of 1,074 s, a model reasoning time of 37.6 s, and 67.3 model parameters, as depicted in Fig. 6. In contrast to other approaches, this article achieves the highest model accuracy alongside the fastest model runtime. LSTM's distinct advantages lie in its adeptness at handling complex, nonlinear, and long sequence data. Conversely, traditional algorithms like KNN, LR, and SVM might struggle to capture temporal correlations within sequence data due to their reliance on distance-based or linear sample relationships. Moreover, while KNN, LR, and SVM are more suitable for linearly separable or straightforwardly structured data, their capacity to model nonlinear data is comparatively weaker.

## DISCUSSION

The construction of ecological environment teaching of new media network teaching is never achieved overnight. It requires efforts from many aspects, starting from the ecological perspective and closely surrounding the niche relationship between teachers and students to ensure the balance between the teacher's teaching student's position and the student's niche. In the context of new media and online teaching, the following aspects can be strengthened (*Daradkeh, 2022*). First, change teachers' teaching ideas. In the open education activities under the background of audio-visual new media, teachers should pay attention to innovation and attract students to actively participate in learning; teachers should also recognize the importance of practice, to design teaching processes, create teaching resources, and cultivate students' practical ability. Secondly, set up teachers and technical teams. If teachers cannot master some production software in a short time, professional teaching teachers and technology application practice teachers can form teaching teams to give play to their respective advantages and cooperate to build open education resources. In short, in the era of new media, we cannot just say nothing but do something. We should combine what we say with what we do. When we strengthen the teaching of new media, we should strengthen it in practice to ensure that teaching is not divorced from reality (*Castañeda & Mauricio, 2020*).

At the same time, in the era of new media, open education, and teaching forms are a bright spot that attracts students. With the continuous development and popularization of internet information, the application (APP) in intelligent terminal devices is increasingly rich and perfect. In software design, it is necessary to meet the needs of group interaction and participation, give full play to the advantages of peer learning, actively guide students to learn independently and cooperatively and encourage them to discuss and compete (*Kim, 2019*). In addition, in the process of designing the APP, we need to consider students with different degrees of knowledge, different majors and different hobbies, and consider the future needs and development potential of students; in the design of teaching content, teaching methods and teaching extension, through the diversified setting of learning paths, it is ensured that each student can use different learning paths to further improve himself through different degrees of learning and exploration while meeting his professional knowledge needs. For English learning in universities, we should improve it in four aspects:

listening, speaking, reading, and writing. In addition to the traditional online teaching, by combining with the model proposed in this article, teachers can better understand students' learning status and better carry out the follow-up design and task arrangement of the course. The LSTM model proposed in this article is relatively simple, but the recognition rate exceeds 96% is the performance shown is superior to the traditional machine learning model. At the same time, with the enhancement of the current chip computing power, this method can be easily transplanted to mobile clients and portable devices to realize the supervision of students' learning status. For example, for some types of students who have too long learning time, reduce the recommendation of corresponding courses through intelligent recommendation and supervision algorithms or improve their comprehensive strength through other types of learning and testing (*Chau, Law & Tang, 2021*). Therefore, the method proposed in this article has a good development prospect in the application of network teaching ecology teaching under the new media platform.

## CONCLUSION

This article investigates the English teaching network ecology under the background of new media and data mining technology and makes a data-based study of the teacher-student niche relationship in the teaching ecology. The data used by learners in the software is collected by peripheral devices of network platform and processed into observed values. The intermediate LSTM method of deep neural network is used to classify the learning sequence of learners. Then, by comparing the recognition results of different hidden layer elements, the optimal model is obtained, and the accuracy rate is more than 96%. In addition, compared with other methods, LSTM has obvious advantages in dealing with complex and non-linear long series data, with faster model running speed and more effective capture of time correlation. It can be found that the proposed model can help teachers better understand students' learning habits, to formulate the next learning plan efficiently and ensure the balance between teacher and student niches. Restricted by the differences between data platform collection and different terminal statistical methods, the method proposed in this article has certain limitations. In the future research, we hope to improve the model by collecting more terminal data, add new model evaluation indicators, and compare different model algorithms, so as to provide support for the construction of China's new media network teaching ecology.

## ACKNOWLEDGEMENTS

We thank the anonymous reviewers whose comments and suggestions helped to improve the manuscript.

### Funding

This work was supported by College of Technology, Hubei Engineering University's "One Excellent Course for One Teacher" Project in 2023-Qualified Course College English (I) (No.12 [2023] of Hugong New Institute of Technology) and the research results of the

2023 teaching research project of College of Technology, Hubei Engineering University-the construction of college English smart classroom from the perspective of artificial intelligence (2023JY15). The funders had a role in the study design. The funders had no role in data collection and analysis, decision to publish, or preparation of the manuscript.

## Grant Disclosures

The following grant information was disclosed by the authors:

College of Technology, Hubei Engineering University's "One Excellent Course for One Teacher" Project in 2023-Qualified Course College English (I) (No.12 [2023] of Hugong New Institute of Technology).

2023 Teaching Research Project of College of Technology, Hubei Engineering University: 2023JY15.

## Competing Interests

Muhammad Aleem is an Academic Editor for PeerJ.

## Author Contributions

- Chen Chen conceived and designed the experiments, analyzed the data, performed the computation work, prepared figures and/or tables, and approved the final draft.
- Muhammad Aleem conceived and designed the experiments, performed the experiments, performed the computation work, authored or reviewed drafts of the article, and approved the final draft.

## Data Availability

The code is available in the Supplemental File. The data is available at Zenodo: Erkan Er. (2020). Student activity data (1.0) [Data set]. Zenodo. https://doi.org/10.5281/zenodo.5059711.

## Supplemental Information

Supplemental information for this article can be found online at http://dx.doi.org/10.7717/peerj-cs.1869#supplemental-information.

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
