# Peer review of "A model for new media data mining and analysis in online English teaching using long short-term memory (LSTM) network"

_PeerJ Computer Science, doi:10.7717/peerj-cs.1869_

## Round 0.1 · original submission · Major Revisions

Please see the detailed reviews. Language clarity, pedagogical implications, data processing insights, comparative analysis, ethical considerations, and visual representation are suggested improvements. Recommendations include clarifying the goal of helping teachers, detailing the data collection process, simplifying data-to-observation explanations, introducing LSTM-based classification, explaining test outcomes, and emphasizing precision's importance for educators. Both reviews stress the need to comprehensively explore the proposed model's impact on learning states, ethical considerations, and the significance of achieved accuracy for future educational applications.

**Language Note:** The review process has identified that the English language must be improved. PeerJ can provide language editing services - please contact us at copyediting@peerj.com for pricing (be sure to provide your manuscript number and title). Alternatively, you should make your own arrangements to improve the language quality and provide details in your response letter. – PeerJ Staff

Reviewer 1 ·

Basic reporting

This paper tries to address learning state recognition that resonates with promise, offering a tempting prospect for solving the tricky tapestry of student learning behaviours. Several facets exist where augmentations could decidedly increase the importance and transparency of your research.

Start by clearly stating the goal—helping teachers understand students' progress. Emphasize how data collected from learners using software aids in this process.
Describe the data collection process more clearly, possibly through some diagram. Explain how various factors like screen time, sensor attitude, and network strength contribute to forming learning observation values.
Simplify the explanation of turning collected data into observations. Describe how these observations are formed from user characteristics and technical factors to give educators a clearer understanding.
Introduce the intelligent classification method for learning time series—highlight the use of Long Short-Term Memory (LSTM) in creating a deep network model. Make it clearer how this model recognizes students' learning status.
Explain the test outcomes more explicitly—how the proposed model achieved over 95% accuracy in recognizing time series. Emphasize the significance of this precision for teachers in understanding students' learning status.
Connect the achieved precision to its importance for future educational applications. Please explain how this high accuracy aids teachers in intelligently understanding students' learning states, making it easier for educators to provide support.

Experimental design

please see the above comments

Validity of the findings

please see the above comments

Additional comments

please see the above comments

Reviewer 2 ·

Basic reporting

Your work evinces excellent promise, and I’m optimistic that these suggested enhancements will significantly augment its contribution to educational technology. I eagerly anticipate witnessing how these refinements further fortify your already compelling research.
(1) Enhancing Clarity in Language: Ensuring linguistic precision and clarity throughout the paper to facilitate a seamless comprehension experience for a diverse readership is paramount.

Experimental design

(2) Articulating Pedagogical Implications: An expansive exploration delineating how this intelligent recognition of learning status can serve as a lodestar for shaping innovative teaching methodologies and fostering improved student outcomes would impart significant depth.

(3) Deepening Discussion on Data Processing Techniques: Offering richer insights into the preprocessing techniques applied before channeling data into the LSTM model would engender a deeper appreciation of the methodology.

(4) Conducting Comparative Analysis: The inclusion of a comparative analysis juxtaposing the proposed model's performance against existing methodologies or models in the field would offer invaluable context and validation.
(5) Ethical Contemplation: Initiating a discourse on ethical considerations or robust data privacy measures implemented during data collection and analysis would enhance the paper's credibility and ethical standing.

(6) Visual Representation Augmentation: Infusing visual aids, such as meticulously crafted charts or illuminating graphs, to visually elucidate data patterns or model performance would enrich the paper's comprehensibility.

Validity of the findings

(8) Exploration of Future Trajectories: Proposing avenues for further research endeavors or contemplating potential enhancements to the model would invigorate scholarly discourse and pave the way for continuous refinement.

---

## Round 0.2 · accepted · Accept

Both reviewers have confirmed that the authors have addressed all of their comments.

Reviewer 1 ·

Basic reporting

changes accepted

Experimental design

changes accepted

Validity of the findings

changes accepted

Additional comments

changes accepted

Reviewer 2 ·

Basic reporting

no comment

Experimental design

no comment

Validity of the findings

no comment